# Inflammation in VTA Caused by HFD Induces Activation of Dopaminergic Neurons Accompanied by Binge-like Eating

**DOI:** 10.3390/nu14183835

**Published:** 2022-09-16

**Authors:** Runan Sun, Mariko Sugiyama, Sixian Wang, Mitsuhiro Kuno, Tomoyuki Sasaki, Tomonori Hirose, Takashi Miyata, Tomoko Kobayashi, Taku Tsunekawa, Takeshi Onoue, Yoshinori Yasuda, Hiroshi Takagi, Daisuke Hagiwara, Shintaro Iwama, Hidetaka Suga, Hiroshi Arima

**Affiliations:** 1Department of Endocrinology and Diabetes, Nagoya University Graduate School of Medicine, Nagoya 466-8550, Japan; 2Research Center of Health, Physical Fitness and Sports, Nagoya University, Nagoya 464-8601, Japan; 3Department of Endocrinology and Diabetes, Ichinomiya Municipal Hospital, Ichinomiya 491-8558, Japan; 4Department of Gastroenterology and Metabolism, Nagoya City University Graduate School of Medical Sciences, Nagoya 467-8602, Japan

**Keywords:** obesity, binge-like eating, high-fat diet, inflammation, reward system, ventral tegmental area, feeding behavior

## Abstract

Binge eating is a characteristic symptom observed in obese individuals that is related to dysfunction of dopaminergic neurons (DNs). Intermittent administration of a high-fat diet (HFD) is reported to induce binge-like eating, but the underlying mechanisms remain unclear. We generated dopaminergic neuron specific IKKβ deficient mice (KO) to examine the effects of inflammation in DNs on binge-like eating under inflammatory conditions associated with HFD. After administration of HFD for 4 weeks, mice were fasted for 24 h, and then the consumption of HFD was measured for 2 h. We also evaluated that the mRNA expressions of inflammatory cytokines, glial markers, and dopamine signaling-related genes in the ventral tegmental area (VTA) and striatum. Moreover, insulin was administered intraventricularly to assess downstream signaling. The consumption of HFD was significantly reduced, and the phosphorylation of AKT in the VTA was significantly increased in female KO compared to wild-type (WT) mice. Analyses of mRNA expressions revealed that DNs activity and inflammation in the VTA were significantly decreased in female KO mice. Thus, our data suggest that HFD-induced inflammation with glial cell activation in the VTA affects DNs function and causes abnormal eating behaviors accompanied by insulin resistance in the VTA of female mice.

## 1. Introduction

Obesity is a chronic disease that occurs when energy intake exceeds energy expenditure, and its prevalence has been increasing in recent decades [1]. Abnormal eating behavior is a known characteristic of obesity and is accompanied by irregular eating, including excessive intake of highly palatable and caloric foods such as those typically seen in a high-fat diet (HFD) [2,3], and binge eating disorders [4]. Binge eating is an abnormal eating behavior that results in a loss of control over the appropriate amount of food intake and is characterized by excess consumption of food in a short period of time [5]. It has been suggested that the cause of these eating behavior abnormalities is a dysfunction of the reward system in the brain [6].

Dopaminergic neurons (DNs) projecting from the ventral tegmental area (VTA) of the midbrain to the striatum play a central role in the regulation of reward system-based eating behavior [7]. Multiple studies have reported that insulin receptor signaling in DNs inhibits hedonic feeding behavior [8]. Nasal administration of insulin in humans has been reported to decrease food taste evaluation and value signaling in mesolimbic regions [9]. Additionally, inflammation is known to be an adverse factor that causes dysfunction of DNs [10,11], and our recent study revealed that HFD causes inflammation and insulin resistance in the VTA [12]. However, it remains unclear whether inflammation in the VTA is associated with how HFD administration affects feeding behavior. 

IKKβ plays a central role in the transduction of inflammatory signals such as tumor necrosis factor-α (TNF-α) [13]. IKKβ also causes insulin resistance by directly phosphorylating serine residues of insulin receptor substrate-1 (IRS-1) [14]. Previous studies have shown that brain or hypothalamic neuron-specific IKKβ deficiency ameliorates obesity induced by HFD administration [15], suggesting that central inflammation induced by HFD plays a pivotal role in the regulation of energy balance. 

In our present study, we evaluated the eating behavior of mice lacking dopaminergic neuron-specific IKKβ (KO) compared to wild-type (WT) mice, using a binge-like eating model under conditions of VTA inflammation induced by HFD administration. 

## 2. Materials and Methods

### 2.1. Mice

All animal procedures were approved by the Animal Care and Use Committee of Nagoya University Graduate School of Medicine and performed in accordance with National Institutes of Health animal care guidelines. Mice were maintained as described previously [16].

### 2.2. Mice with Dopamine Transporter (DAT)-Specific Deletion of IKKβ

*IKKβ^lox/lox^* mice were provided by EMMA (RRID: IMSR_EM:01921). *DAT-Cre* trans-gene mice (RRID: IMSR_JAX:006660) express functional Cre-recombinase only in DNs [17]. DNA extraction and genotyping were performed as described previously [16]. Primer sequences used for genotyping of *IKKβ^lox/lox^* and *DAT-Cre* mice were as follows: *IKK2* forward, 5′ACAGGCTGCCAGTTAGGGAGGAAG; reverse, 5′-GGAGTACTGCCAAGGAGGAGAT; *DAT-Cre* forward, 5′-TGGCTGTTGGTGTAAAGTGG; reverse, 5′-GGACAGGGACATGGTTGACT (to detect WT gene) or 5′- CCAAAAGACGGCAATATGGT (to detect transgene). All *IKKβ^lox/lox^* mice, *DAT-Cre* mice were backcrossed more than 15 generations onto a C57BL/6J background.

### 2.3. Isolating DNA from Tissues for Detection of Recombination of Floxed Alleles

DNA was extracted from different tissues (VTA, substantia nigra, hypothalamus, cerebral cortex, hippocampus, cerebellum, brain stem) of mice at the age of 10 weeks, and genotyping was performed as described previously [18].

### 2.4. Body Composition and Food Intake

At weaning (3 weeks of age), mice were placed on HFD (Test Diet 58Y1, PMI Nutrition International, KS, USA; 60.9% fat,18.3% protein, and 20.1% carbohydrate). Body weight was monitored until 16 weeks of age. Measurements of epididymal fat pad weight, perigonadal fat pad weight, and blood glucose were performed at 16 weeks of age in the beginning of the light cycle (between 09:00 and 09:30 a.m.) when mice were in the fed state. Food intake of HFD was assessed by multifeeders (Shinfactory, Fukuoka, Japan) at 16 weeks of age. Feed efficiency was calculated as grams of body weight gained per grams of food consumed over a 3-day period.

### 2.5. Assessment of Feeding Behaviors under Fast-Refeed Access to HFD or Chow Diet (CD)

Mice (6 weeks of age) were divided into 2 groups. After acclimation, WT and KO mice were exposed to HFD for 4 weeks, and then were fasted for 24 h. After fasting, the mice were divided into two groups: “HFD group” and control diet “CD group” on the experimental day. The mice in the HFD group were given restricted access to HFD for 120 min (21:00 to 23:00), while the mice in the CD group were given restricted access to CD (CE-2, CLEA Japan, Tokyo, Japan; 4.6% fat, 24.9% protein and 70.5% carbohydrate). Then, HFD and CD intakes were measured for 120 min (21:00 to 23:00). The food intake of mice on both the CD and HFD were assessed by multifeeders (Shinfactory, Fukuoka, Japan). Mice were sacrificed at 120 min after the start of access to HFD or CD (23:00).

### 2.6. Extraction of Brain Tissues

After mice were sacrificed, the VTA, NAc, and CPu were quickly dissected as described previously [16]. Then the dissected tissues were immediately frozen in liquid nitrogen until RNA extraction.

### 2.7. Determination of mRNA Levels by qRT-PCR

Total RNA was extracted from samples, and copy DNA was synthesized as described previously [16]. Quantitative reverse transcriptase (qRT)-PCR reactions were carried out as described previously [16]. The relative mRNA levels of *TNF-α, IL1β* (interleukin-1β)*, PTP1B* (Protein tyrosine phosphatase 1B)*, Socs3* (Suppressor of cytokine signaling 3)*, IL10* (interleukin-10)*, Iba1* (ionized calcium binding adaptor molecule 1)*, CD11b* (cluster of differentiation molecule 11B)*, Emr1* (EGF-like module-containing mucin-like hormone receptor-like 1)*, CD68* (cluster of differentiation 68)*, GFAP* (glial fibrillary acidic protein), *GLAST* (glutamate aspartate transporter)*,*
*cFos* (cellular oncogene fos)*, ΔfosB* (FBJ murine osteosarcoma viral oncogene homolog B)*, DAT, TH* (Tyrosine hydroxylase)*, D1R* (Dopamine receptor D_1_), and *D2**R* (Dopamine receptor D_2_) were assessed by qRT-PCR as described previously [16]. *Gapdh* (glyceraldehyde 3-phosphate dehydrogenase) was used as an internal control. The sequences of primers are described in Appendix A.

### 2.8. Intracerebroventricular Injection of Insulin

After overnight fasting, 10 weeks-old mice on HFD were deeply anesthetized as described previously [12]. After anesthetization, insulin (10^−5^ M) or saline in a volume of 2.0 μL was injected into the lateral ventricle as described previously [12]. Then, 15 min after injection, the mice were decapitated and dissected the VTAs. They were stored at −80 °C until analysis. 

### 2.9. Determination of Protein Levels by Western Blot

Western blotting was performed using proteins extracted from VTA as described previously [12]. The antibodies used to assess Akt phosphorylation were the same as previously reported [12].

### 2.10. Immunohistochemistry

Brain collection for immunohistochemistry was performed as described previously [16]. The immunohistochemistry was performed using free-floating method described as previously [16]. The used antibodies were rat anti-DAT (MAB369, Merck KGaA, Darmstadt, Germany, RRID: AB_2190413), and rabbit anti-IKKβ (#2684, Cell Signaling Technology, Danvers, MA, USA, RRID: AB_2122298).

### 2.11. Statistical Analysis

The statistical analysis was performed using SPSS Statistics 27 (IBM, Endicott, NY, USA; RRID:SCR_002865) described as previously [16]. Results are expressed as mean ± standard error of the mean (SEM), and differences were considered significant at *p* < 0.05.

## 3. Results

### 3.1. Generation of Dopaminergic Neuron-Specific IKKβ Deficient Mice

To generate dopaminergic neuron-specific IKKβ deficient mice, we crossed *IKKβ^lox/lox^* mice with *DAT-Cre* heterozygous mice to generate *IKKβ^lox/lox^*
*DAT-Cre* mice (KO mice) and *IKKβ^lox/lox^* littermate controls (WT mice). Deletion of the *IKK2* allele in KO mice was only detected in DNA extracts from the VTA (Figure 1A). In contrast, no recombined alleles were detected in WT mice (Figure 1A). IKKβ immunostaining revealed that IKKβ was expressed in the DNs of the VTA in WT mice but was rarely expressed in KO mice (Figure 1B).

### 3.2. IKKβ Signaling in DNs Does Not Affect Energy Balance and Glucose Metabolism

We examined the role of IKKβ signaling in the VTA in the regulation of energy balance and glucose metabolism under ad libitum access to HFD for 13 weeks in male (WT = 14, KO = 14) and female (WT = 12, KO = 6) mice. There were no significant differences in body weight (Figure 2A,B), daily food intake (Figure 2C,E), feed efficiency (Δ body weight/Δ food intake) (Figure 2D,F), epididymal fat pad weight (Figure 2G), or perigonadal fat pad weight (Figure 2I) between WT and KO mice on HFD. There were no significant differences between genotypes in glucose metabolism estimated by blood glucose (Figure 2H,J). These data suggest that IKKβ signaling in the VTA does not affect energy balance or glucose metabolism under ad libitum access to HFD.

### 3.3. IKKβ Deficiency in DNs Suppresses Binge Eating of HFD in Female Mice

#### 3.3.1. Binge-like Eating under Inflammatory Conditions in the VTA

Our previous study showed that HFD feeding for 4 weeks causes inflammation and insulin resistance in the VTA [12]. To investigate the hedonic regulation of HFD intake under inflammatory conditions in the VTA, we performed an experimental protocol (Figure 3) based on a previous study [19] with some modifications. In the protocol, WT and KO mice were fasted for 1 day after exposure to HFD for 4 weeks, and the cumulative refeeding consumption of HFD or CD was measured at 30, 60, and 120 min on the experimental day (21:00 to 23:00). Binge eating refers to eating large amounts of food in a short period of time [5]. According to a previous study [20], binge eating was defined as 25% or more of total daily calorie intake in one hour, and we confirmed that the refeeding consumption of HFD at 60 min in both male and female WT mice was more than 25% of the total daily calorie intake (data not shown). We also confirmed that the stomach of the mouse after refeeding of HFD for 120 min was extremely expanded (Appendix A).

#### 3.3.2. IKKβ Deficiency in DNs Suppresses Binge Eating of HFD in Female Mice

There were no significant differences in the cumulative refeeding consumption of HFD (WT = 6, KO = 8) or CD (WT = 8, KO = 9) between genotypes in male mice (Figure 4A,C). In contrast, in female KO mice, the cumulative refeeding consumption of HFD (WT = 13, KO = 11) was significantly decreased at all time points (*p* < 0.05) (Figure 4B) compared to female WT mice, and the amount of HFD intake at 60 min was less than 25% of the total daily calorie intake (data not shown). The cumulative refeeding consumption of CD (WT = 12, KO = 10) was significantly increased at 60 and 120 min in female KO mice compared to WT mice (*p* < 0.05) (Figure 4D), but the amount of CD intake at 60 min did not reach 25% of the total daily calorie intake (data not shown). In addition, in male mice, the phosphorylation of AKT induced by intracerebroventricular injection of insulin after exposure to HFD for 4 weeks followed by fasting for 1 day was significantly decreased in HFD compared to CD in both WT (CD = 7, HFD = 6, *p* < 0.05) and KO (CD = 6, HFD = 6, *p* < 0.05) mice (Figure 4E). In comparison, in female mice, the AKT phosphorylation was significantly decreased on HFD compared to CD in WT mice (CD = 5, HFD = 6, *p* < 0.05), whereas there were no significant differences between HFD and CD conditions in KO mice (CD = 6, HFD = 6) (Figure 4F).

### 3.4. IKKβ Deficiency in the Dopamine Neurons of VTA Suppresses HFD Induced Inflammation in the VTA of Female Mice

As our previous study showed that HFD feeding for 4 weeks causes inflammation in the VTA [12], we next evaluated the expressions of inflammatory-related molecules in the VTA of WT and KO mice. In male mice (WT = 8, KO = 8), only Iba1 mRNA expression levels were significantly decreased in KO mice compared to WT mice (*p* < 0.05) (Figure 5A,C). In comparison, in female mice (WT = 7, KO = 7), the mRNA expression levels of TNF-α, PTP1B, Iba1, CD11b, and GFAP in the VTA were significantly decreased in KO mice compared to WT mice (*p* < 0.05) (Figure 5B,D). The mRNA expression levels of the anti-inflammatory cytokine IL10 were significantly increased in KO mice compared to WT (*p* < 0.05). These data suggest that the suppression of VTA inflammation associated with IKKβ deficiency is more robust in females than in males, and that inflammation in the VTA is associated with glial cell activation in female mice.

### 3.5. IKKβ Deficiency in the Dopamine Neurons of VTA Altered the Dopamine-Related Gene Expressions in Female Mice

Given that female KO mice showed suppression of binge eating accompanied by improved insulin resistance and reduced inflammation in the VTA under HFD conditions (Figure 4B,F and Figure 5B), we next assessed the activity of DNs and dopamine signaling by evaluating the dopamine-related gene expressions of WT (male = 8, female = 8) and KO (male = 8, female = 8) mice in the VTA, NAc, and CPu. In the VTA, only DAT mRNA expression levels showed significant decreases in both male and female KO mice compared to WT mice (*p* < 0.05) (Figure 6A,B). In the NAc, there were no significant differences of DAT, cFos, ΔFosB, D1R, or D2R mRNA expression levels between genotypes (Figure 6C,D). In the CPu, both male and female mice showed significant decreases in cFos mRNA expression levels in KO mice compared to WT mice (*p* < 0.05). Significant decreases in DAT, ΔFosB, D1R, and D2R mRNA expression levels were detected in female KO mice (*p* < 0.05) but not in male KO mice compared to WT mice (Figure 6E,F). These data suggest that the suppression of binge eating observed in female KO mice is associated with suppression of dopaminergic neuronal activity and signaling in the CPu.

## 4. Discussion

This study investigated the effect of inflammation in VTA on eating behavior under HFD conditions. Our findings demonstrate that IKKβ deficiency in VTA DNs (i) suppresses binge eating of HFD in female mice; (ii) suppresses HFD-induced inflammation accompanied by suppression of glial cell activation in the VTA of female mice; (iii) improves insulin resistance induced by HFD in the VTA of female mice; (iv) alters the dopamine-related gene expressions in female mice; and (v) does not affect energy balance. These findings suggest that HFD-induced inflammation with glial cell activation in the VTA affects dopaminergic neural function and causes abnormal eating behavior accompanied by insulin resistance in the VTA of female mice independent of energy balance.

Overeating is a characteristic eating behavior observed in obese patients, and this abnormal eating behavior is thought to be due to a dysfunction of DNs in the mesolimbic system [21,22]. There are several lines of evidence that inflammation occurs in reward-related brain regions with the use of abused substances [10]. For example, alcohol and cocaine exposure both cause inflammation in the VTA [23,24]. We previously showed that HFD caused inflammation with microglial activation and insulin resistance in the VTA of mice [12], but whether these conditions actually affected feeding behavior remained unclear. Here, we show that insulin resistance accompanied by HFD-induced inflammation in the VTA promotes binge eating. Female KO mice had improved VTA insulin resistance associated with HFD administration, resulting in significantly reduced food intake in the binge-eating model compared to WT. The phenotype observed in female KO mice was consistent with a previous report showing that insulin action in the VTA suppresses ingestive behaviors, such as the hedonic feeding, food craving, and salience of food cues [8]. Furthermore, male KO mice showed less pronounced suppression of VTA inflammation by HFD, and inflammation was essentially the same between KO and WT mice. As a result, insulin sensitivity in the VTA was not significantly different, and food intake in the binge-eating model was not significantly different between genotypes either. These data indicate that inflammation and insulin resistance in the VTA caused by HFD are essential for binge eating.

Our present study suggests that suppression of inflammatory signaling in DNs suppresses glial cell activation associated with HFD administration, which is consistent with previous reports [25,26]. Protein Tyrosine Phosphatase 1B (PTP1B) expression was decreased in female KO mice in which VTA inflammation was suppressed. PTP1B is classically known as an enzyme that inhibits insulin receptor signaling [27]; we and others have reported that PTP1B expression is enhanced by inflammation [28]. We have also previously reported that PTP1B deficiency suppresses hypothalamic inflammation associated with HFD [29], which is accompanied by enhanced expression of IL10, an anti-inflammatory cytokine. Consistent with this previous report, we observed enhanced IL10 expression in the VTA of female KO mice in our present study, suggesting that the reduction of PTP1B expression exerts an anti-inflammatory effect as well as improves insulin receptor signaling in the VTA. 

In binge-eating models, it has been reported that intermittent administration of sucrose or HFD enhances DAT expression in the VTA, suggesting that dopamine reuptake is involved in binge eating [20,30]. In our present study, DAT expression was decreased by suppression of IKKβ signaling in the VTA, and DAT expression was markedly suppressed in female mice in which the suppression of inflammation was more pronounced. Although the regulatory mechanism of DAT expression in the VTA is still unclear, the presence or absence of inflammation may be critical for DAT expression under the inflammatory conditions associated with HFD administration. We have previously shown that intermittent HFD administration enhances dopamine signaling in the CPu [16]. In the present study, we showed that D1R and D2R expression was decreased and the expression of downstream signals, cFos and ΔFosB, was attenuated in the CPu of female KO mice, indicating that the enhancement of dopamine signaling upon HFD administration was suppressed. In addition, female KO mice in which DNs inflammation was suppressed ate less HFD, but more normal food, suggesting that preference for palatable food was attenuated. These results are consistent with a previous report using a resistance to binge eating model showing that intake of control chow food was increased in comparison to a highly palatable food [31].

The limitation of this experiment is that it is unclear why there was a sex difference in feeding behavior. According to previous reports, estrogen is known to suppress inflammation [32,33]. Combined with the fact that estrogen receptors are present in DNs [34] and in the VTA [35], it is possible that the suppression of inflammation associated with IKKB deficiency in female mice may have been further amplified by the presence of estrogen. In addition, dopamine release was not directly measured in this study, and it remains unclear whether the decreased dopamine signaling in CPu is due to decreased dopamine levels.

## 5. Conclusions

A graphical abstract of the present study is shown in Figure 7. In conclusion, our data suggest that inflammation and insulin resistance in the VTA caused by HFD lead to activation of the DNs and binge-like eating of HFD. Our findings also suggest that inflammation in the VTA represents a new target of drug discovery for binge-like eating and obesity.

## Figures and Tables

**Figure 1 nutrients-14-03835-f001:**
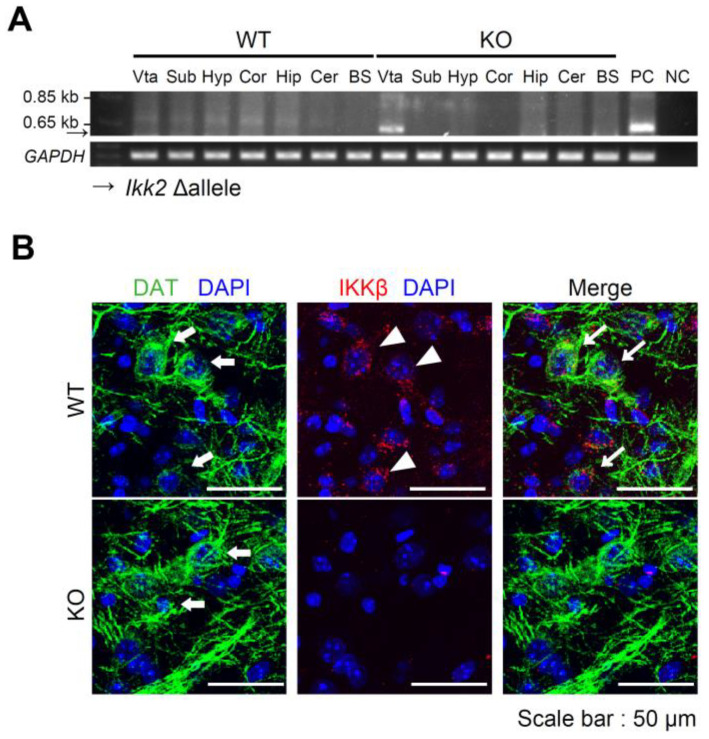
Generation of dopaminergic neuron-specific IKKβ deficient mice. (**A**) Detection of deleted *IKK2* alleles (Δ) in *IKKβ^lox/lox^*
*DAT-Cre* (KO) mice. DNA was extracted from different tissues as follows, ventral tegmental area (Vta); substantia nigra (Sub); hypothalamus (Hyp); cerebral cortex (Cor); hippocampus (Hip); cerebellum (Cer); brain stem (BS); positive control (PC); negative control (NC). The deletion of the floxed allele was detected by PCR. GAPDH was used as an internal control. (**B**) Representative pictures showing DAT (green), IKKβ (red), and DAPI (blue) staining in the VTA of WT and KO mice. White thick arrow heads indicate co-localization of DAT and DAPI; white triangles indicate co-localization of IKKβ and DAPI; white thin arrow heads indicate co-localization of DAT, DAPI and IKKβ.

**Figure 2 nutrients-14-03835-f002:**
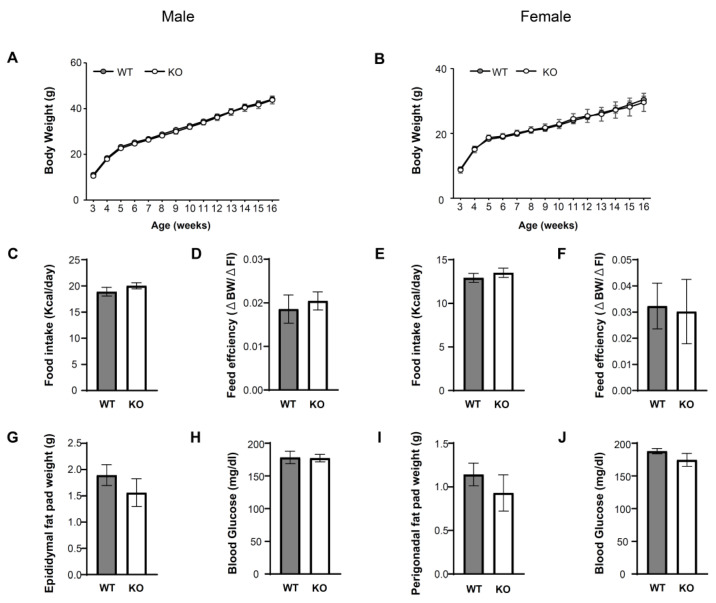
There were no statistically significant differences between WT and KO mice in terms of energy balance after 13 weeks HFD administration. (**A**,**B**) Body weight of male (**A**) and female (**B**) *IKKβ^lox/lox^* (WT) and *IKKβ^lox/lox^*
*DAT-Cre* (KO) mice on HFD for 13 weeks. (**C**,**E**) Daily food intake of male (**C**) and female (**E**) 16-week-old WT and KO mice (**D**,**F**) feed efficiency of male (**D**) and female (**F**) 16-week-old WT and KO mice (**G**,**I**) epididymal fat pad weight (**G**) and perigonadal fat pad weight (**I**) 16-week-old WT and KO mice (**H**,**J**). Blood glucose of male (**H**) and female (**J**) WT and KO mice at the age of 16 weeks. Data are mean ± SEM. Two-way ANOVA with repeated measures was used for statistical analysis of A and B, or an unpaired t test was used for statistical analysis of C to J. The details of statistics are shown in Appendix A.

**Figure 3 nutrients-14-03835-f003:**
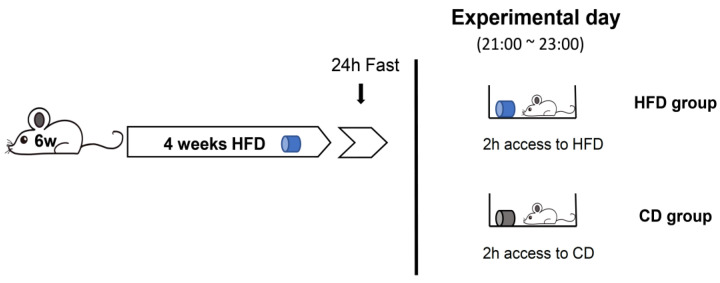
Refeeding of HFD after 24 h fasting gives rise to binge-like eating. The protocol for the experiment. WT and KO mice were exposed to a HFD for 4 weeks, and then were fasted for 24 h. On the day of the experiment, the mice were divided into two groups, “HFD group” and “CD group”. The HFD and CD groups were given HFD and CD, respectively, for 2 h.

**Figure 4 nutrients-14-03835-f004:**
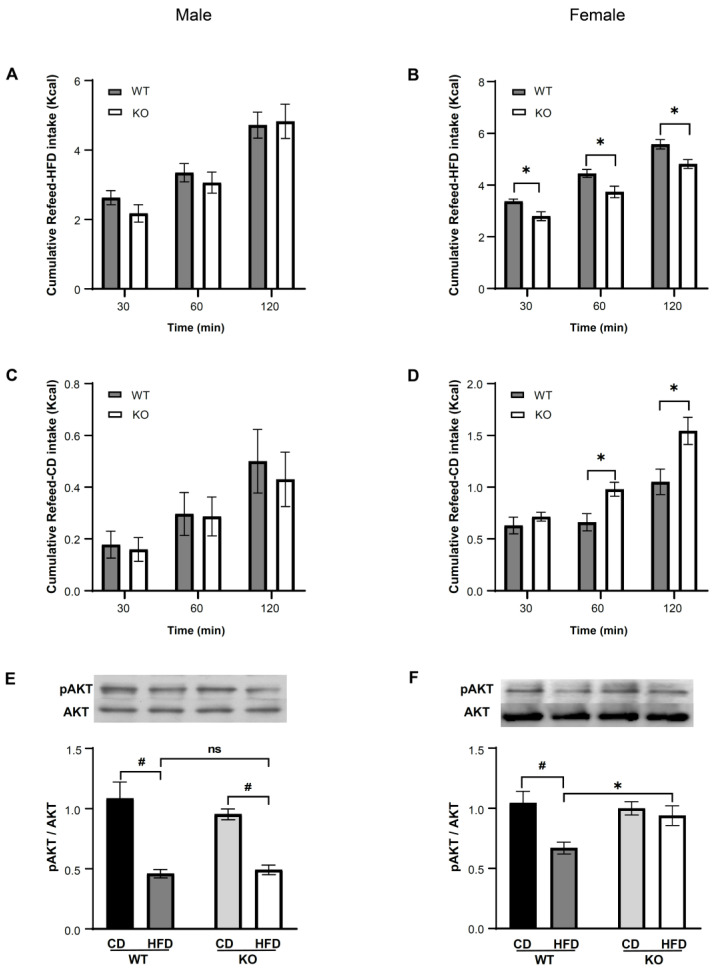
IKKβ deficiency in the dopamine neurons of VTA suppresses the refeeding consumption of HFD after 24 h fasting and improves insulin resistance in female mice on HFD for 4 weeks. (**A**,**B**) The refeeding-HFD food intake over 120 min of male (**A**) and female (**B**) WT and KO mice after 24 h fasting on HFD for 4 weeks. (**C**,**D**) The refeeding-CD food intake over 120 min of male (**C**) and female (**D**) WT and KO mice after 24 h fasting on HFD for 4 weeks. (**E**,**F**) Phosphorylation of AKT 15 min after insulin injection in male (**E**) and female (**F**) WT, KO mice on CD and HFD for 4 weeks. Data are mean ± SEM. Two-way ANOVA assessed with repeated measures was used for statistical analysis of A to D. Two-way factorial ANOVA was used for statistical analysis of E and F. * *p* < 0.05, versus WT; # *p* < 0.05, versus CD; ns: no statistical significance. The details of statistics are shown in Appendix A. Uncropped images of Western blots used in this analysis are shown in Appendix A.

**Figure 5 nutrients-14-03835-f005:**
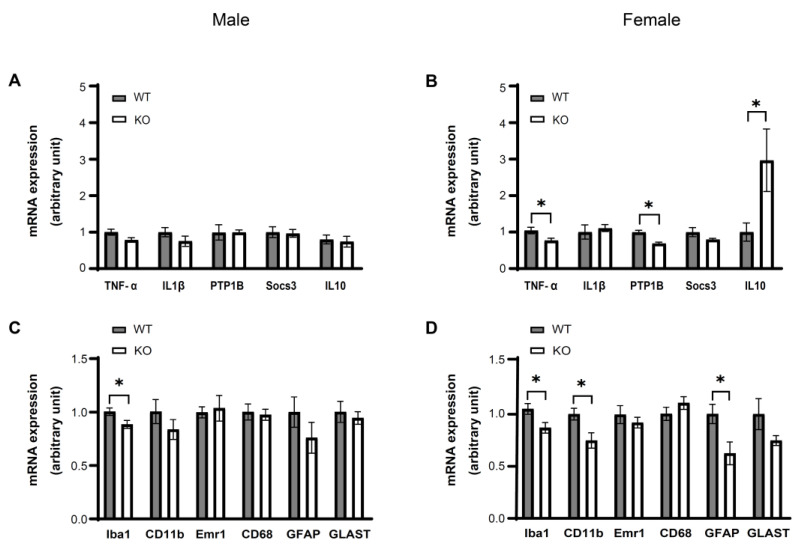
IKKβ deficiency in the dopamine neurons of VTA suppresses inflammation in the VTA caused by HFD feeding for 4 weeks in mice. (**A**–**D**) The *TNF-α, IL1β, PTP1B,*
*S**ocs3, IL10* (**A**,**B**) and *Iba1, CD11b, Emr1, CD68, GFAP, GLAST* (**C**,**D**) mRNA expression levels of male (**A**,**C**) and female (**B**,**D**) WT and KO mice on HFD for 4 weeks. Data are mean ± SEM. Unpaired t test was used for statistical analysis of A to D. * *p* < 0.05, versus WT. The details of statistics are shown in Appendix A.

**Figure 6 nutrients-14-03835-f006:**
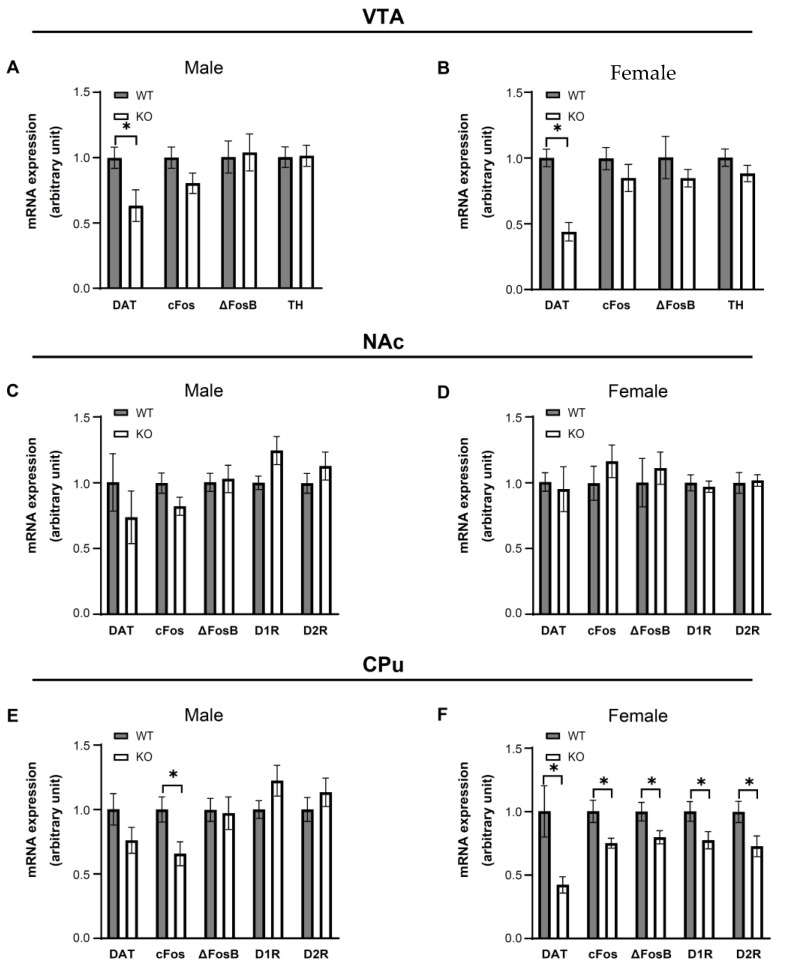
HFD feeding for 4 weeks alters dopamine-related gene expressions in mice. (**A**–**F**) Gene expressions were measured in the VTA (**A**,**B**), NAc (**C**,**D**), and CPu (**E**,**F**) of male (**A**,**C**,**E**) and female (**B**,**D**,**F**) WT and KO mice on HFD for 4 weeks. Data are mean ± SEM. Unpaired t test was used for statistical analysis of A to F. * *p* < 0.05, versus WT. The details of statistics are shown in Appendix A.

**Figure 7 nutrients-14-03835-f007:**
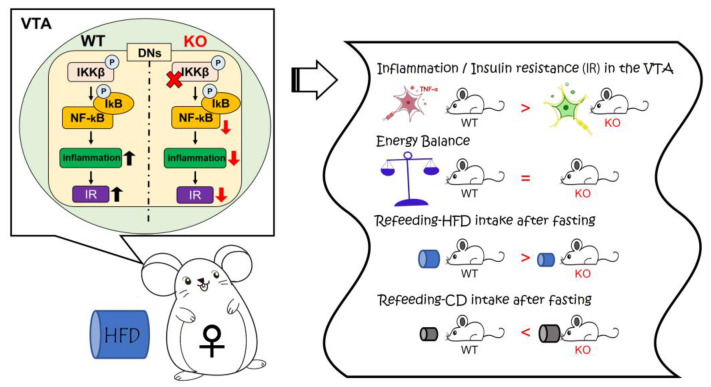
Schematic illustration of the regulation of feeding behaviors by IKKβ signaling in the VTA. IR: insulin resistance; DNs: dopamine neurons.

## Data Availability

The data presented here are available on request from the corresponding author.

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
