# Peer review of "Inflammation in VTA Caused by HFD Induces Activation of Dopaminergic Neurons Accompanied by Binge-like Eating"

_nutrients, 2022, doi:10.3390/nu14183835_

Round 1

Reviewer 1 Report

Interesting manuscript that aims to: “To evaluate the eating behavior of mice lacking dopaminergic neuron-specific IKKβ (KO) compared to wild-type (WT) mice, using a binge-like eating model under conditions of VTA inflammation induced by HFD administration ”

The results are part of the group's line of research, and although the results are very interesting, I think it would be good to hypothesize how these results could be extrapolated to humans, especially to high-fat diets.

Regarding the conclusion, it is not written, it only mentions “A graphical abstract of the present study is shown in Figure 7. Our findings revealed that inflammatory signaling in dopamine neurons plays an important role in the control of eating behaviors associated with high-fat diet administration.

Author Response

Interesting manuscript that aims to: “To evaluate the eating behavior of mice lacking dopaminergic neuron-specific IKKβ (KO) compared to wild-type (WT) mice, using a binge-like eating model under conditions of VTA inflammation induced by HFD administration”

The results are part of the group's line of research, and although the results are very interesting, I think it would be good to hypothesize how these results could be extrapolated to humans, especially to high-fat diets.

Response:

According to the suggestion, we have added additional commentary in the Discussion and Conclusion section of the revised manuscript (page 12: line 394-395; page 13: line 440-442).

Regarding the conclusion, it is not written, it only mentions “A graphical abstract of the present study is shown in Figure 7. Our findings revealed that inflammatory signaling in dopamine neurons plays an important role in the control of eating behaviors associated with high-fat diet administration.

Response:

The conclusion has been modified accordingly (page 13: line 438-440).

Reviewer 2 Report

This manuscript describes several experiments to investigate the role of inflammation in the VTA in a high-fat diet mouse model of obesity. It addresses both chronic food intake and acute food intake. Well done.

Major comments: None

Minor comments:

1. Please provide a detailed macronutrient breakdown for the control diet (section 2.4). The HFD is already described.

2. Page 4. Consider moving "insulin was 10 -5 M" from line 162 to line 155, after "10ul syringe"

3. Please add an indication of n numbers to all methods and results. I see these are described very well in the supplementary tables, but I think n numbers would help increase the impact of the manuscript if they are included in the description of the methods and results.

4. Please add P values to the results section throughout. 

5. Page 8 line 287. The cut off of 25% seems arbitrary. Please provide a rationale with reference, or justify why 25% is chosen.

End

Author Response

This manuscript describes several experiments to investigate the role of inflammation in the VTA in a high-fat diet mouse model of obesity. It addresses both chronic food intake and acute food intake. Well done.

Major comments: None

Minor comments:

1. Please provide a detailed macronutrient breakdown for the control diet (section 2.4).  The HFD is already described.

Response:

  According to this suggestion, we have provided a detailed macronutrient breakdown for the control diet in section 2.5. of the revised manuscript (page 3: line 117-118).

2. Page 4. Consider moving "insulin was 10 -5 M" from line 162 to line 155, after "10ul syringe"

Response:

We have moved the description of insulin to after the description of the syringe in section 2.8. of the revised manuscript (page 4: line 158-159).

3. Please add an indication of n numbers to all methods and results. I see these are described very well in the supplementary tables, but I think n numbers would help increase the impact of the manuscript if they are included in the description of the methods and results.

Response:

We have added n numbers in the Materials and Methods and in the Results sections of the revised manuscript (page 6: line 244-245; page 8, line 290, 292, 295, 300-301,302-303, 304; page 10, line 324,326,346-347).

4. Please add P values to the results section throughout.

Response:

We have added P values in the Results section of the revised manuscript (page 8: line 292, 296, 300, 301, 303; page 10, line 325, 328, 330, 349; page 11, line 353, 354).

5. Page 8 line 287. The cut off of 25% seems arbitrary. Please provide a rationale with reference, or justify why 25% is chosen.

Response:

The rationale of the cut off value of 25% has been described in more detail in the revised manuscript (page 7: line 272-274).